# Shifting from Left Ventricular Ejection Fraction to Strain Imaging in Aortic Stenosis

**DOI:** 10.3390/diagnostics13101756

**Published:** 2023-05-16

**Authors:** Vasileios Anastasiou, Stylianos Daios, Maria-Anna Bazmpani, Dimitrios V. Moysidis, Thomas Zegkos, Theodoros Karamitsos, Antonios Ziakas, Vasileios Kamperidis

**Affiliations:** First Department of Cardiology, AHEPA Hospital, Medical School, Aristotle University of Thessaloniki, 54636 Thessaloniki, Greece; vasianas44@gmail.com (V.A.); stylianoschrys.daios@gmail.com (S.D.); mariannabaz@hotmail.gr (M.-A.B.); dimoysidis@gmail.com (D.V.M.); zegkosth@gmail.com (T.Z.); karamits@gmail.com (T.K.); tonyziakas@hotmail.com (A.Z.)

**Keywords:** aortic stenosis, ventricular damage, left ventricular ejection fraction, strain imaging, echocardiography, cardiac computed tomography, cardiac magnetic resonance

## Abstract

Adverse ventricular remodeling is an inflexion point of disease progression in aortic stenosis (AS) and a major determinant of prognosis. Intervention before irreversible myocardial damage is of paramount importance to sustain favorable post-operative outcomes. Current guidelines recommend a left ventricular ejection fraction (LVEF)-based strategy to determine the threshold for intervention in AS. However, LVEF has several pitfalls: it denotes the left ventricular cavity volumetric changes and it is not suited to detecting subtle signs of myocardial damage. Strain has emerged as a contemporary imaging biomarker that describes intramyocardial contractile force, providing information on subclinical myocardial dysfunction due to fibrosis. A large body of evidence advocates its use to determine the switch from adaptive to maladaptive myocardial changes in AS, and to refine thresholds for intervention. Although mainly studied in echocardiography, studies exploring the role of strain in multi-detector row computed tomography and cardiac magnetic resonance are emerging. This review, therefore, summarizes contemporary evidence on the role of LVEF and strain imaging in AS prognosis, aiming to move from an LVEF-based to a strain-based approach for risk stratification and therapeutic decision-making in AS.

## 1. Introduction

Aortic stenosis (AS) poses pressure overload to the myocardium, causing ventricular damage that depends, among other things, on the afterload excess, and the period of exposure. In order to restore wall stress and maintain cardiac output under those circumstances, the myocardium will initially respond with compensatory hypertrophy of the left ventricle (LV), defined as excess mass, increase in relative wall thickness, and concentric hypertrophy [1]. LV hypertrophy leads to impaired compliance and elevated end-diastolic pressures [2]. If AS is left untreated, LV interstitial space expands and diffuse myocardial fibrosis develops, followed by replacement fibrosis and cardiomyocyte death at a later stage [3]. This marks an inflexion point of irreversible myocardial damage and the transition to a decompensating phase, which translates to macroscopic impairment of the LV systolic properties [4]. From this point, the development of heart failure symptoms and malignant arrhythmias is common, dramatically increasing the risk of mortal events (Figure 1).

It is therefore imperative that intervention on the valve is implemented at an appropriate stage of the disease, in order to avoid irreversible myocardial damage and its consequences. While the guidelines recommend left ventricular ejection fraction (LVEF) as the gold standard to describe myocardial damage and proceed to therapeutic maneuvers [5,6], a large body of evidence has demonstrated the superior prognostic role of deformation strain imaging. This review aims to aggregate and comprehensively appraise the current literature addressing the role of LVEF and strain imaging, by echocardiography, cardiac computed tomography, and cardiac magnetic resonance, in AS, and seeks to explain the rationale behind a possible LV strain imaging-based strategy to determine treatment decisions.

## 2. Echocardiography for Left Ventricular Function

### 2.1. Prognostic Value of Left Ventricular Ejection Fraction

Symptoms serve as a sign of established disease and dismal prognosis, and should prompt intervention, irrespective of LV function, in high-gradient severe AS, as long as the patient is not critically ill or overly fragile, for them to benefit from intervention [6]. However, cautious assessment of LV systolic function is of paramount importance in case the patient is free of symptoms, and both ESC and ACC/AHA guidelines recommend intervention in asymptomatic severe AS, with evidence of systolic LV dysfunction attributable to the valve disease [5,6]. Guidelines suggest the use of LVEF to define systolic dysfunction. More specifically, intervention is advocated with Class I indication when the LVEF is <50%, and with IIa when the LVEF is <55% by ESC [6]. Going even further, the AHA/ACC suggest considering the threshold of 60% when a progressive decline in at least three serial imaging studies has been documented [5].

In a retrospective analysis of 2017 symptomatic and asymptomatic AS patients undergoing surgical aortic valve replacement (AVR), Dahl et al. demonstrated a stepwise increase in all-cause mortality with lower preoperative LVEF values [7]. The group with preoperative LVEF < 50% suffered the worst outcome, whereas asymptomatic patients with LVEF ≥ 60% had the lowest mortality rate [7]. In keeping with those findings, another study demonstrated that patients with severe symptomatic and asymptomatic AS and LVEF < 50% had the worst cumulative five-year incidence of death and heart failure hospitalization, irrespective of treatment strategy (conservative or initial AVR) [8]. Similarly, severe LV dysfunction, expressed as LVEF < 30%, was associated with higher mortality rates among patients undergoing transcatheter aortic valve replacement (TAVR) [9].

A few studies have endeavored to identify an association between higher LVEF thresholds and outcomes. Capoulade et al. studied a large symptomatic and asymptomatic AS population, with at least mild AS, and concluded that patients with LVEF < 55% and stroke volume index < 35 mL/m^2^ had the worst survival rates [10]. Although LVEF was an independent predictor of all-cause mortality for the entire population (*p* < 0.001), it lost this association (*p* = 0.56) when subgroup analysis was performed for asymptomatic patients [10]. On the contrary, Bohbot et al. demonstrated that LVEF < 55% was a strong, independent predictor of excess mortality in a large cohort of asymptomatic or minimally symptomatic patients with severe AS and LVEF > 50% [11]. Patients treated both medically and surgically with LVEF < 55% had the worst prognosis, as compared with subjects with LVEF > 55% [11]. Going even higher in the range of LVEF, values < 60% were identified as an independent predictor of all-cause mortality in asymptomatic severe AS subjects [12]. Ito et al. demonstrated that an LVEF < 60%, at the time of moderate AS, could predict further deterioration of LVEF, so that many patients will already have LVEF < 50% by the time of initial severe AS diagnosis [13]. In this regard, the authors raised the question of early intervention of moderate AS, when LVEF is less than 60%, to prevent future decline in myocardial function [13]. Those studies exemplified that clinically relevant LV systolic dysfunction may already be established when LVEF is 50% to 59%, questioning the guideline-recommended thresholds for intervention [6].

It should be highlighted that a drop in LVEF < 50%, due to severe AS, is almost invariably accompanied by symptoms, and the prevalence of asymptomatic severe AS with LVEF < 50% is only 0.4% [14]. Hence, a strategy of waiting for LVEF to fall to <50% to decide on intervention, in the absence of symptoms, seems suboptimal to sustain good post-operative results, and the most recently established threshold of 55% holds promise to optimize outcomes [6]. Moreover, LVEF has several recognized limitations and masks disease progression in cases of pronounced LV remodeling and small LV cavity. In this setting, LVEF will tend to increase in parallel with the extent of concentric remodeling, rather reflecting the relationship of wall thickness to cavity size [15]. Thus, in paradoxical low-flow, low-gradient AS, a decrease in cardiac output, SV, and myocardial contractility may occur, despite a preserved LVEF [16].

Nonetheless, LVEF remains the most widely available assessor of LV systolic function that still plays an important role in assessment of AS. Details of the large scale studies addressing the prognostic role of preoperative LVEF in different AS populations are presented in Table 1.

### 2.2. Prognostic Value of Left Ventricular Global Longitudinal Strain

In spite of its widespread use, LVEF remains an oversimplified measure of assessing myocardial function, as it only describes the volumetric alterations of the LV cavity. Beyond LVEF, more contemporary measures of subclinical LV dysfunction have entered the clinical arena, allowing for earlier recognition of ventricular damage. Speckle-tracking echocardiography represents a method of quantifying myocardial deformation and aims to capture early decline of intrinsic myocardial contractile function, before LVEF impairment is manifested (Figure 2) [20]. Global longitudinal strain (GLS) is the most established application of speckle-tracking echocardiography. This technique is based on detecting and following the movement of myocardial speckles in the longitudinal axis. It is considered less angle- and geometry-dependent, less preload- and afterload-dependent [21,22], and more reproducible than LVEF [23,24].

Longitudinal subendocardial fibers are the most susceptible to increased wall stress and reduced perfusion, and will be the first to be impaired by the afterload excess. GLS is capable of highlighting such subtle myocardial changes at the subendocardial level before a drop in LVEF occurs [24,25]. Yingchoncharoen et al. studied 79 asymptomatic patients with severe AS and preserved LVEF, and concluded that GLS retained its independent association with cardiovascular outcomes, even after correction for clinical and echocardiographic parameters in different models [26]. Those findings were confirmed in a larger cohort of 395 asymptomatic AS patients with preserved LVEF, where the subset with the lowest LV GLS (<−12.1%) displayed the worst survival [27]. Another study focused on the role of basal longitudinal function in asymptomatic AS subjects and proved that it could predict future AVR [28]. A pathophysiological explanation is that myocardial diffuse interstitial fibrosis and focal mid-wall fibrosis starts from the basal parts of the ventricle in AS, which can be indirectly detected by GLS [29]. A recent meta-analysis of asymptomatic AS patients with preserved LVEF demonstrated that a GLS of −14.7% was the best cut-off for death prediction and it was associated with a >2.5 increment of mortality [30]. This body of evidence begins to establish the role of GLS as a guide for AVR in asymptomatic AS patients at an earlier phase than indicated by LVEF. The results of the Danish National Randomized Study on Early Aortic Valve Replacement in Patients With Asymptomatic Severe Aortic Stenosis (DANAVR) trial (NCT03972644) are currently awaited to determine if asymptomatic patients with abnormal LVGL Sby echocardiography will benefit from early AVR.

GLS is an invaluable marker of LV dysfunction in paradoxical low flow, low gradient (LFLG) AS with prominent concentric remodeling and small cavity size, where LVEF will be supranormal and not representative of LV function. This was indicated by Kamperidis et al. in a group of 134 paradoxical LFLGAS patients, who had worse all-cause mortality with GLS > −15% [31], and was confirmed by other investigators [32]. In the case of classical LFLG AS, reduced LVEF is the sequela of long-standing disease and is invariably accompanied by impaired GLS. Even in this subset of patients, GLS can elicit significant incremental prognostic information beyond that obtained with LVEF [33]. Rest GLS and stress GLS, obtained during low-dose dobutamine stress echocardiography, were independent predictors of mortality in a large cohort of classical LFLG AS patients [34].

After the intervention, if the valve-related pressure overload has been retracted from the ventricle on time, regression of diffuse fibrosis and myocardial cellular hypertrophy may occur. In this instance, GLS is a sensitive marker of myocardial systolic recovery, while LVEF might fail to detect such subtle functional improvement [35,36]. It has been proposed that GLS > −13.3% predicts a lack of myocardial recovery after TAVR [36], while the magnitude of improvement in GLS has been shown to have a prognostic impact on survival after TAVR [37]. In LFLG AS, GLS improved one year after TAVR, in both classical and paradoxical type groups, whereas LVEF failed to identify the LV functional recovery in the paradoxical type group [38]. For the patients with classical LFLG AS, although TAVR is considered the preferred treatment choice a baseline, GLS may be able to determine the TAVR-responders; GLS > −12% has been suggested for identifying patients with a lack of flow reserve during dobutamine stress echocardiography and lack of reverse remodeling after TAVR [39]. The main features of studies addressing the prognostic role of GLS by echocardiography are presented in Table 2.

### 2.3. Prognostic Role of Non-Invasively Assessed Left Ventricular Myocardial Work

Despite being a sensitive surrogate of occult longitudinal systolic dysfunction, GLS still does not account for the global afterload which differs with AS severity and peripheral vascular resistance. Global afterload is the cause of increased metabolic demand and oxygen consumption of cardiomyocytes in order to maintain cardiac output. Russel et al. described a novel non-invasive method that reflects those metabolic demands by assessing different parameters of the energy efficiency of the LV [49,50]. This method estimates LV myocardial work, via dedicated software, by integrating LV longitudinal strain measurements by speckle-tracking echocardiography, LV afterload estimates, and cardiac event timing to derive a pressure strain loop. Noninvasive LV pressure curves are formulated according to the duration of ejection and isovolumic phases, as determined by left-sided valve timing events. Four different indices of myocardial work are calculated including:(i)LV global work index, representing the total work within the LV pressure-strain loops;(ii)LV global constructive work, defined as the work performed during myocardial shortening in systole and the work during myocardial lengthening in isovolumic relaxation;(iii)LV global wasted work, representing the work contributing to the lengthening of the cardiac myocytes during systole and the shortening during isovolumic relaxation; and(iv)LV global work efficiency, defined as the percentage of effectively spent work by the LV myocytes, and obtained by the following formula:
$$\mathrm{LV}\;\mathrm{global}\;\mathrm{work}\;\mathrm{efficiency}=\left(\frac{\mathrm{global}\;\mathrm{constructive}\;\mathrm{work}}{\left[\mathrm{global}\;\mathrm{constructive}\;\mathrm{work}\;+\;\mathrm{global}\;\mathrm{wasted}\;\mathrm{work}\right]}\right)\times 100\%$$

Whereas normally non-invasive brachial blood pressure is entered in the module as the equivalent of the LV afterload, this is not the case for severe AS. In severe AS, global afterload is the composite of the peripheral vascular component (brachial blood pressure) and the valvular component (transvalvular mean gradient) posed to the ventricle. Taking this into account, Fortuni et al. developed a method to evaluate LV myocardial work in patients with severe AS, by adding the echocardiography-derived transvalvular mean gradient to the systolic blood pressure [51]. Their formula showed excellent agreement with invasively-derived LV pressure recordings, confirming the accuracy of the proposed method to evaluate myocardial work by echocardiography in severe AS [51]. Global constructive work (odds ratio, 0.941 [95% CI, 0.887–0.998], *p* = 0.042, per 100 mmHg/% increase) and global work index (odds ratio, 0.912 [95% CI, 0.849–0.980], *p* = 0.012, per 100 mmHg/% increase), in a cohort of symptomatic severe AS patients, were the only echocardiography indices independently associated with heart failure symptoms, after adjustment for right ventricular free wall strain [51]. Accordingly, myocardial work is significantly reduced after TAVR, reflecting the valve-related pressure overload withdrawal from the ventricle [52]. The myocardial work parameters seem to be early markers of afterload-related LV maladaptation and functional impairment. Their prognostic validity warrants further exploration in order to assess whether they can add prognostic information beyond that obtained by GLS (Figure 2).

## 3. Multi-Detector Row Computed Tomography for Left Ventricular Function

Computed tomography plays a prominent role in AS assessment, mainly by obtaining information regarding the extent of aortic calcification, aortic valve–aortic root anatomic relationship, and aortic valve area. It is the guideline-recommended modality for establishing severity in paradoxical LFLG AS, and in classical LFLG when there is no flow reserve by dobutamine stress echocardiography [6]. Additionally, it is part of the standard of care in pre-TAVR evaluation to plan the intervention. Recent advances in technology have allowed multi-detector row computed tomography (MDCT) to expand its role by providing information on the myocardial function.

MDCT is the cardiac imaging modality that provides three-dimensional data of the heart with the best spatial resolution, and allows the volumetric quantification of each cardiac chamber and computation of the LVEF and right ventricular EF. Thus, a MDCT-derived staging system of extra-aortic cardiac damage has recently been validated in a population of 405 patients undergoing TAVR [53]. The population was classified into five different stages of disease progression, based on volumetric chamber quantification, LV hypertrophy, and grading of mitral annular calcification acquired entirely from the pre-TAVR MDCT scans [53]. The staging system classified patients as: no myocardial damage (Stage 0), LV damage; LVEF < 50% or LV mass index > 79.2 gr/m^2^ for male and >63.8 gr/m^2^ for female (Stage 1), left atrial or mitral damage; left atrial volume index > 56 mL/m^2^, atrial fibrillation or severe mitral annular calcification (Stage 2), right atrial damage; right atrial volume index > 70 mL/m^2^ (Stage 3) and right ventricular damage; right ventricular EF < 35% (Stage 4) [53]. This system enabled refined risk stratification of severe AS, with Stage 3 and 4 being independently associated with higher all-cause mortality [53], and the results were similar to those obtained by an echocardiography-based staging system [54].

Apart from the volumetric assessment of cardiac function with LVEF and right ventricular EF evaluation, a novel dynamic feature-tracking software has arisen, permitting GLS analysis from MDCT data. Hence, the emerging technology allows shifting from LVEF to LV GLS in MDCT, in parallel with the echocardiography shift from LVEF to LV GLS. MDCT-derived GLS showed excellent correlation with speckle-tracking-echocardiography-derived GLS (r = 0.791, *p* < 0.001) in a cohort of 214 TAVR recipients [55]. Moreover, MDCT-derived GLS, at the pre-TAVR assessment, has emerged as an independent associate of all-cause mortality [56,57], with a cut-off of −14% being proposed to determine the worst outcomes [57]. In a large cohort of 432 symptomatic severe AS patients undergoing TAVR, not only was pre-operative MDCT-derived GLS an independent predictor of the outcome, but also, subjects who exhibited improvement of GLS one month post-TAVR demonstrated favorable clinical outcomes [58]. This finding outlines the capacity of MDCT-derived LV GLS to detect subtle myocardial improvement post-TAVR (Table 3) [58]. Accordingly, it could be beneficial to measure GLS at part of the pre-TAVR MDCT protocol assessment, as it provides valuable information on post-TAVR prognosis [56,57].

## 4. Cardiac Magnetic Resonance for Left Ventricular Function

Cardiac magnetic resonance (CMR) can extend beyond volumetric quantification and has the unique strength of myocardial tissue characterization. The extent of myocardial damage can be effectively quantified in the form of local fibrosis by late gadolinium enhancement and in the form of diffuse fibrosis by T1 mapping.

The distribution of gadolinium corresponds to areas of focal replacement fibrosis, which, histologically, represents irreversible myocardial damage and cardiomyocyte death [61]. Dweck et al. demonstrated that mid-wall late gadolinium enhancement was an independent mortality predictor among symptomatic and asymptomatic patients with at least moderate AS, and provided incremental prognostic information on top of LVEF [62]. Barone-Rockette et al. studied 154 patients with severe symptomatic AS, undergoing surgical AVR, and concluded that the presence of late gadolinium enhancement conveyed poor post-operative survival outcomes [63]. In a multicenter registry of 674 patients with severe AS, who had undergone CMR and were treated with surgical or transcatheter AVR, Musa et al. confirmed the aforementioned results by demonstrating an independent association of late gadolinium presence with post-intervention mortality, irrespective of scar pattern [64].

Myocardial T1 mapping expresses the degree of extracellular volume expansion and diffuse myocardial fibrosis, which presents earlier than focal replacement fibrosis and is reversible if a well-timed AVR is pursued [65]. Higher native T1 values were associated with a higher risk of pre- and post-operative events in a group of 127 symptomatic and asymptomatic patients with at least moderate AS [66]. Extracellular volume fraction represents another T1 mapping-derived method to quantify diffuse myocardial fibrosis. When assessed in 440 patients with severe symptomatic AS awaiting AVR, extracellular volume fraction retained an independent association with mortality, even after correction for late gadolinium enhancement [67]. Both late gadolinium enhancement and T1-mapping provide complementary information regarding myocardial fibrosis and can be used in conjunction to risk stratify AS patients before intervention [68].

Nonetheless, further software development has arisen which enables LV deformation assessment by feature-tracking CMR strain analysis in AS, which can be derived from steady-state free precession cine images [69]. Feature-tracking CMR strain parameters in the longitudinal, radial, and circumferential orientation are significantly reduced in patients with AS, compared to normal subjects [70], even when measured post-TAVR [71]. In a group of 63 severe AS patients treated with surgical AVR, both radial, circumferential and longitudinal tissue-tracking stain parameters showed correlation with LV mass regression and extracellular volume, post-intervention [72]. CMR-derived GLS was an independent predictor of reverse remodeling [72]. With respect to the different AS subgroups, feature tracking GLS showed a mild reduction in high-flow high-gradient AS and a severe reduction in classical LFLG AS, but was preserved in paradoxical LFLG subtype, compared to healthy controls in a small series [73]. In this study, paradoxical LFLG was the only subgroup that did not exhibit an increase of CMR-GLS after TAVR, but the small study cohort was a major limitation [73]. Prognostic information was delivered by a larger cohort of at least moderate AS patients, where a feature tracking CMR-derived GLS values of >−17.9% identified patients with the worst event-free survival [59]. Similarly, a circumferential strain of >−18.7% was associated with reduced survival in a cohort undergoing surgical or transcatheter AVR [74]. In the study by Fukui et al., CMR-derived LV GLS emerged as an independent predictor of all-cause mortality in a group of 123 patients with low-gradient severe AS [60]. In the same study, CMR LV GLS <−11% was used, together with the presence of late gadolinium enhancement and extracellular volume >28%, as part of a three-component CMR risk marker tool, where each component had a cumulative effect on the prognosis [60]. Details of those studies are summarized in Table 3. Ultimately, feature-tracing-based CMR strain analysis may be a useful alternative to echocardiography when local expertise is available, but further proof of its prognostic relevance is necessary to establish its role in AS management.

## 5. Conclusions

LVEF assessed by echocardiography is the main parameter for LV function evaluation in severe AS, endorsed by the current guidelines. However, LVEF is limited to a mere description of the LV volumetric changes during a cardiac cycle. GLS by speckle-tracking echocardiography has emerged as a more sensitive index of LV function, detecting even subtle LV intramyocardial changes that the volumetric LVEF may miss. Considering that LV dysfunction is used as a criterion for AVR, the evaluation of GLS may define the precise time-point for intervention, which is of paramount importance for prognosis, post-AVR. Evidence that GLS may be evaluated with feature-tracking MDCT and CMR has emerged, but the available data are still scarce and further research is warranted to determine its value in clinical practice.

## Figures and Tables

**Figure 1 diagnostics-13-01756-f001:**
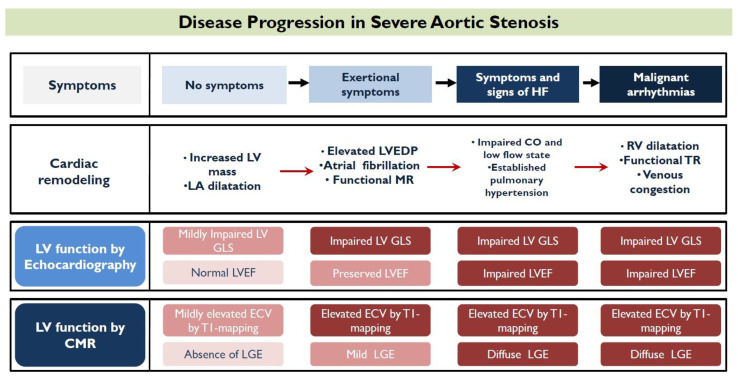
Symptom occurrence is an inflexion point in the progression of disease in aortic stenosis. Figure 1 illustrates, step by step, the evolution of symptoms, the respective progression of ventricular remodeling, and its expression by the main contemporary imaging biomarkers.

**Figure 2 diagnostics-13-01756-f002:**
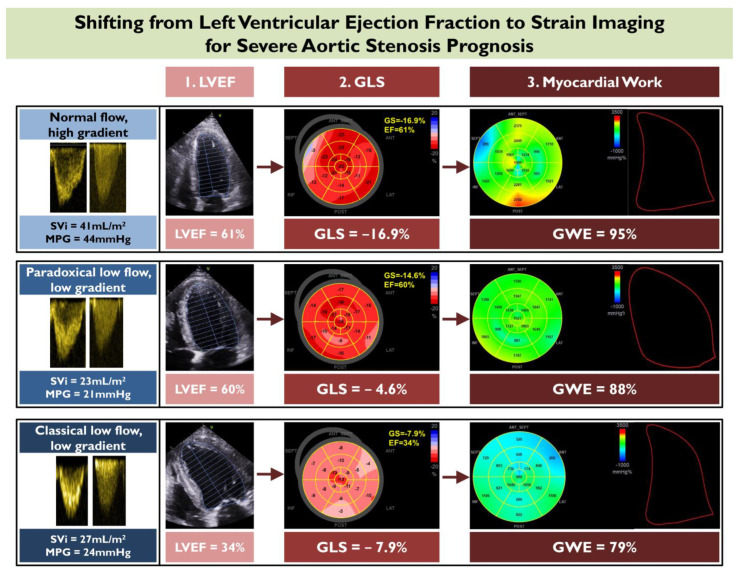
Shifting from left ventricular ejection fraction (LVEF) to strain imaging for severe aortic stenosis prognosis. This figure depicts the progression of three different echocardiographic modalities for assessing myocardial function in aortic stenosis, starting from LVEF and moving to the more contemporary imaging biomarkers of global longitudinal strain (GLS), and myocardial work. An example of applying these modalities is illustrated in three different subgroups of aortic stenosis patients including (i) normal flow, high gradient, (ii) paradoxical low flow, low gradient, and (iii) classical low flow, low gradient. GWE—global work efficiency; MPG—mean gradient; Svi—stroke volume indexed.

**Table 1 diagnostics-13-01756-t001:** Large-scale studies addressing the prognostic role of preoperative left ventricular ejection fraction in aortic stenosis.

1st Author	Year	Patients *n*	AS Population	Intervention	Cut-Off LVEF (%)	Outcomes
Mihaljevic et al. [17]	2008	3049	Severe AS	SAVR	40%	Worst long-term survival (5.1 ± 3.2 years follow up)
Halkos et al. [18]	2008	779	Undergoing SAVR	SAVR	40%	Worst unadjusted 1-, 3-, and 5-year survival rates
Goldberg et al. [19]	2013	5277	Severe AS	SAVR	50%	Worst survival at 6 months and 8 years
Dahl et al. [7]	2015	2017	Symptomatic and asymptomatic severe AS	SAVR	50%	Worst 5-year all-cause mortality (HR, 0.41; 95% CI, 0.35–0.47)
Baron et al. [9]	2016	11,292	Undergoing TAVR	TAVR	30%	Higher 1-year mortality rates compared to 50–50% and >50%
Capoulade et al. [10]	2016	1065	At least mild AS	SAVR	55%	Best cut-off value to predict all-cause mortality
Ito et al. [13]	2018	928	Severe AS with available echo before diagnosis	Not specified	50%	Worse survival compared to LVEF 50–60% and >60%
Taniguchi et al. [8]	2018	3794	Severe AS	SAVR or TAVR	50%	Highest 5-year incidence of composite death or HF hospitalization
Lancellotti et al. [12]	2018	1375	At least moderate asymptomatic AS	SAVR or TAVR	60%	Independent predictor of all-cause mortality [HR: 5.01, (95% CI):(2.93–8.57)]
Bohbot et al. [11]	2019	1678	Asymptomatic or mildly symptomatic severe AS	SAVR	55%	2-fold increase in all-cause mortality

Abbreviations: AS = aortic stenosis; CI = confidence interval; HF = heart failure; HR = hazard ratio; LVEF = left ventricular ejection fraction; SAVR = surgical aortic valve replacement; TAVR = transcatheter aortic valve replacement.

**Table 2 diagnostics-13-01756-t002:** Studies addressing the prognostic role of left ventricular global longitudinal strain, derived by speckle tracking echocardiography in aortic stenosis.

1st Author	Year	*n*	AS Population	Cut-Off LV GLS (-%)	Association with Outcomes
Lancellottiet et al. [40]	2010	163	Moderate and severe AS	15.9	Significant predictive power for MACE
Zito et al. [41]	2011	52	Asymptomatic severe AS	18	Significant predictive power for MACE
Dahl et al. [42]	2012	125	Symptomatic severe AS with LVEF > 40%	10.3	Increased overall mortality, cardiac mortality, and MACEs
Kearney et al. [43]	2012	146	Mild, moderate, and severe AS	15	One-year MACE-free survival was only 25%
Yingchoncharoen et al. [26]	2012	79	Asymptomatic severe AS with LVEF > 50%	15	16% survival rate at 40 months follow up
Kempny et al. [36]	2013	101	Severe AS undergoing TAVR	13.3	Predictor of lack of longitudinal strain recovery post TAVR
Logstrup et al. [37]	2013	100	Severe AS undergoing TAVR	11.95	Pre-TAVR cut-off did not impact on prognosis. LV GLS improvement post TAVR predicted outcomes.
Kamperidis et al. [31]	2014	134	Symptomatic paradoxical low-flow, low-gradient AS with AVAi ≤ 0.6 cm^2^/m^2^	15	Mortality rate of 22.4% 3 years after AVR
Kusunose et al. [27]	2014	395	Moderate and severe AS with LVEF > 50%	12.1	43% death during 4.4 ± 1.4 years of follow up
Sato et al. [32]	2014	98	Paradoxical low-flow, low-gradient AS with AVAi ≤ 0.6 cm^2^/m^2^	17	MACE-free survival rate at 2-year follow-up was 57.5%
Nagata et al. [44]	2015	104	Asymptomatic severe AS with LVEF > 50%	17	Significant predictive power for MACE
Dahou et al. [34]	2015	126	Low-flow, low-gradient AS with LVEF ≤ 40% and AVAi ≤ 0.6 cm^2^/m^2^	9	49% 3-year survival
Suzuki Eguchi et al. [45]	2018	128	Severe AS undergoing TAVR	10.6	Freedom from events for patients with GLS ≤ −10.6% occurred more often compared to GLS > −10.6
Vollema et al. [25]	2018	220	Asymptomatic severe AS	18.2	Higher risk for symptoms development or requiring aortic valve intervention
D’Andrea A et al. [39]	2019	75	Classical low-flow, low-gradient severe AS undergoing TAVR	12	Identified patients with lack of reverse remodeling after TAVR
Povlsen et al. [46]	2020	411	Severe AS undergoing TAVR	14	Independent predictor of all-cause mortality
Fukui et al. [47]	2020	510	Symtomatic severe AS	16	Patients with normal LVEFs but reduced GLS had worst survival that those with normal LVEFs and reduced GLS
Lee et al. [48]	2022	412	Severe AS undergoing TAVR	16	Independent predictor of all-cause death and the composite outcome

Abbreviations: AS—aortic stenosis; AVAi—aortic valve area index; AVR—aortic valve replacement; HR—hazard ratio; LVEF—left ventricular ejection fraction; LV GLS—left ventricular global longitudinal strain; MACE—major adverse cardiovascular event; TAVR—transcatheter aortic valve replacement.

**Table 3 diagnostics-13-01756-t003:** Studies addressing the prognostic role of left ventricular global longitudinal strain derived by multi-detector row computed tomography and cardiac magnetic resonance in aortic stenosis.

1st Author	Year	*n*	AS Population	Cut-Off LV GLS (-%)	Association with Outcomes
**Multi-detector row computed tomography**					
Fukui et al. [56]	2020	223	Severe AS undergoing TAVR	20.5	Independent association with all-cause mortality and composite outcome
Gegenava et al. [57]	2020	214	Severe AS undergoing TAVR	14	After 48 months of follow-up, rate of all-cause mortality for GLS ≤ −14% was 15%, versus 28% for GLS > −14%
Fukui et al. [58]	2022	431	Severe AS undergoing TAVR	18.2	GLS > −18.2% had a higher risk of the composite outcome than GLS ≤ −18.2% (HR, 1.77; 95% CI, 1.18–2.66; *p* = 0.006)
**Cardiac magnetic resonance**					
Kim et al. [59]	2020	123	Asymptomatic moderate to severe AS with preserved LVEF	17.9	GLS > −17.9% had worse event-free survival than GLS < −17.9%
Fukui et el. [60]	2022	147	Low-gradient moderate to severe AS	12.4	GLS < −12.4% was associated with a higher risk for all-cause mortality and composite outcome

Abbreviations: AS—aortic stenosis; AVAi—aortic valve area index; CI—confidence interval; HR—hazard ratio; LVEF—left ventricular ejection fraction; LV GLS—left ventricular global longitudinal strain; TAVR—transcatheter aortic valve replacement.

## Data Availability

Not applicable.

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
