# Peer review of "Shifting from Left Ventricular Ejection Fraction to Strain Imaging in Aortic Stenosis"

_diagnostics, 2023, doi:10.3390/diagnostics13101756_

Round 1

Reviewer 1 Report

The review discusses the importance of identifying adverse ventricular remodeling in aortic valve stenosis as a critical point in disease progression. It highlights the need for intervention before fibrosis and irreversible myocardial damage occurs to achieve better post-operative outcomes. The current guidelines rely on left ventricular ejection fraction as the only measure of left ventricular function to determine the intervention threshold in aortic valve stenosis. However, the authors argue that left ventricular ejection fraction has limitations since it primarily reflects changes in ventricular cavity volume and may not detect subtle signs of myocardial damage. To address these limitations, the authors propose the use of strain as a modern imaging tool. A growing body of evidence supports the use of strain imaging to determine the transition from adaptive to maladaptive myocardial changes, thus refining the thresholds for surgical intervention in valvular heart disease. While most studies have focused on echocardiography, there are emerging studies exploring the role of strain in multi-detector row computed tomography and cardiac magnetic resonance. 

Overall, the review emphasizes the limitations of left ventricular ejection fraction and highlights the potential of strain imaging as a more sensitive and accurate method for detecting myocardial damage in aortic valve stenosis. It suggests that a shift towards a strain-based approach could lead to improved risk assessment and better treatment decisions for patients with aortic valve stenosis.

Author Response

We thank the reviewer for the very positive appraisal and for not requesting any changes.

Reviewer 2 Report

The manuscript entitled ‘Shifting from Left Ventricular Ejection Fraction to Strain Imaging in Aortic Stenosis’ provides a well-structured and comprehensive literature overview of the recent studies that empower the strain-based approach for determining the risk stratification and the threshold for intervention in aortic stenosis.

Minor comments:

Table 1 –  may more suitable to use Outcomes instead of Association with events

All tables: please insert the abbreviations as table footnotes

Page 8: ‘Prognostic role of Left Ventricular non-invasive myocardial work’ – please reformulate (non-invasively assessed LV myocardial work)

Minor editing of English language required

Author Response

We thank the reviewer for the positive appraisal.

Reply to Minor Comments:

  1. Table 1 has been edited accordingly –  Outcomes is used instead of Association with events
  2. In all tables the title is placed above the Table and the abbreviations below the Table, as footnotes
  3. In Page 8, the ‘Prognostic role of Left Ventricular non-invasive myocardial work’ has been rephrased and writen as "Prognostic role of non-invasively assessed left ventricular myocardial work"
  4. Minor eidting of the English language has been made